# PLA2R antibody, PLA2R rs4664308 polymorphism and PLA2R mRNA levels in Tunisian patients with primary membranous nephritis

Tarak Dhaouadi[1]*, Jihen Abdellatif[1], Raja Trabelsi[2,3], Hanene Gaied[2,3], Sameh Chamkhi[1], Imen Sfar[1], Rym Goucha[2,3], Fethi Ben Hamida[2,3], Taieb Ben Abdallah[1,2], Yousr Gorgi[1]

1 Research Laboratory in Immunology of Renal Transplantation and Immunopathology (LR03SP01), Charles Nicolle Hospital, Tunis El Manar University, Tunis, Tunisia, 2 Department of Nephrology and Internal Medicine, Charles Nicolle Hospital, Tunis, Tunisia, 3 Research Laboratory of Kidney Diseases (LR00SP01), Charles Nicolle Hospital, Tunis, Tunisia

* dhaouaditarak@yahoo.fr

**Data Availability Statement:** All relevant data are within the paper and its Supporting Information files.

## Abstract

### Background

Primary membranous nephritis (PMN) is an autoimmune disease induced by the deposit of antibodies (Ab) to the phospholipase receptor A2 receptor (PLA2R) on podocytes. In this context, we aimed to assess the relationships between anti-PLA2R Ab, PLA2R rs4664308 SNP, PLA2R mRNA levels and PMN susceptibility and outcome.

### Methods

Sixty-eight PMN patients, 30 systemic lupus erythematosus (SLE) patients with secondary MN and 30 healthy control subjects served for anti-PLA2R Ab measurement by ELISA and PLA2R rs4664308 SNP genotyping by a commercial real-time PCR. Twenty patients with tubulo-interstitial nephritis (TIN) were used as controls for renal PLA2R mRNA quantification in PMN patients from kidney biopsies. PLA2R mRNA quantification was carried-out by real-time PCR after RNA extraction.

### Results

Forty-three (63.2%) PMN patients received initial therapy consisting of alternating monthly cycles of corticosteroids and cyclophosphamide. Twelve (17.6%) patients had resistant PMN to initial therapy and were consecutively treated by cyclosporine or tacrolimus.

Anti-PLA2R Ab were positive in 54 (79.4%) PMN patients, while all SLE patients and controls were negative, $p < 0.0001$. Moreover, anti-PLA2R Ab levels were significantly higher in PMN patients (134.85 [41.25–256.97] RU/ml) than in SLE patients (3.35 [2.3–4.35] RU/ml) and controls (2 [2–2.3]), $p < 0.0001$. Consequently, a ROC curve showed for 100% specificity a sensitivity of 94.1% at a threshold of 2.6 RU/ml. Besides, Anti-PLA2R antibodies levels were significantly associated to non-remission; $p = 0.002$.

**Funding:** This study was supported by the Research Laboratory in Immunology of Renal Transplantation and Immunopathology (LR03SP01), and the Research Laboratory of Kidney Diseases (LR00SP01), Charles Nicolle Hospital, Tunis El Manar University, Tunisia.

**Competing interests:** The authors have declared that no competing interests exist.

The rs4664308*A wild-type allele was significantly more frequent in PMN patients (0.809) than in controls (0.633) and SLE patients (0.65); $p = 0.008$, OR [95% CI] = 2.44 [1.24–4.82] and $p = 0.016$, OR [95% CI] = 2.27 [1.15–4.5], respectively.

Renal PLA2R mRNA levels were significantly higher in PMN patients (218.29 [66.05–486.07]) than in TIN patients (22.09 [13.62–43.34]), $p<0.0001$. Moreover, PLA2R mRNA levels were significantly higher in non-remission patients (fold-factor vs. partial remission = 2.46 and fold-factor vs. complete remission = 12.25); $p = 1.56\ 10E\text{-}8$. In addition, PLA2R mRNA and anti-PLA2R Ab levels were significantly correlated, Spearman Rho = 0.958, $p<0.0001$.

## Conclusion

Anti-PLA2R Ab and renal PLA2R mRNA could be useful markers for PMN outcome predicting. The PLA2R rs6446308 SNP is associated with PMN susceptibility in Tunisians.

## Introduction

Membranous nephropathy (MN) is the most frequent etiology of adult-onset nephrotic syndrome [1]. Twenty percent of MN are secondary membranous nephropathy (SMN) which can be associated to an autoimmune disease such as systemic lupus erythematosus (SLE) or caused by infections, drugs and cancers [2]. The remaining 80% of MN are considered idiopathic or primary membranous nephropathy (PMN) [2].

The discovery in 2009 of the phospholipase A2 receptor (PLA2R) as a target autoantigen by Beck et al [3] has revolutionized the understanding of the pathogenic mechanisms of PMN. In fact, the majority of patients had serum anti-PLA2R antibodies (Ab) [3]. Of note, anti-PLA2R Ab are an heterogenous population of autoantibodies which mainly recognize the immunodominant domain, the cysteine-rich domain (CysR) [4]. These autoantibodies were also found in the membrane deposits and colocalized with the PLA2R in podocytes [3]. Afterwards, several published studies confirmed this initial finding. Thus, screening for these anti-PLA2R antibodies has become an essential component in the diagnostic process for PMN.

In 2010, Liu et al [4] reported an association between the rs35771982 SNP located at exon 5 of the PLA2R gene and the susceptibility to PMN. Similarly, a Korean study [5] noted the association of rs35771982 and rs3828323 SNPs with PMN risk. Moreover, in a study including 3 European cohorts [6], the rs4664308*A wild-type allele had the most significant association among the PLA2R SNPs; $p = 8,6\ 10E\text{-}29$, OR [95% CI] = 2,28 [1,96–2,64]. However, genetic studies of PLA2R to date have not been detailed enough to determine how the genetic risk results in a pathological mechanism that causes the disease [7]. It remains unclear which polymorphisms are responsible and how they affect the PLA2R expression and immunogenicity [7]. Interestingly, the rs4664308 (+1624 A>G) SNP with the smallest $p$-value is in strong linkage disequilibrium with the rs3749117 SNP ($r2 = 0.70$) [6]. Of note, this rs3749117 encodes a nonsynonymous amino acid substitution (M292V) in the extracellular C-type lectin domain 1 (CTLD1) of the PLA2R [6] which could decrease its immunogenicity and may partly explain the rs4664308*G allele protective role. Nevertheless, the linkage disequilibrium between two or more alleles varies ethnically and needs to be assessed in independent cohorts.

Some authors hypothesized that a renal overexpression of PLA2R could trigger the immune response and be correlated to anti-PLA2R Ab production [7]. Indeed, studies of renal PLA2R

expression by immunohistochemistry and direct immunofluorescence have shown a significant increase in the membrane deposits of PLA2R in patients with PMN comparatively to patients with SMN or tubular nephropathies [7,8]. Nevertheless, it remains unknown whether this enhanced staining is the result of an increase in PLA2R gene transcription or not. Of note, in the study of Hoxha et al [8], the authors did not report any significant difference in PLA2R expression between patients with enhanced PLA2R staining and those with no enhanced staining.

Therefore, we aimed in the present study to evaluate the diagnostic and prognostic performances of anti-PLA2R Ab measurement, to investigate the impact of the rs4664308 SNP of PLA2R on PMN susceptibility and outcome, and to examine the renal PLA2R expression in order to look for a possible correlation with the anti-PLA2R Ab production and the rs4664308 polymorphism.

## Material and methods

### Subjects

This study included 68 PMN patients, 30 SLE patients and 30 healthy voluntary blood donors from the same ethnic origin (Tunisian). Twenty patients with tubulointerstitial nephritis (TIN), established by histology, were used as controls for PLA2R mRNA analysis.

A total of 68 patients with biopsy-proven PMN were prospectively investigated between June 2017 and April 2019 in the nephrology and internal medicine department of Charles Nicolle Hospital in Tunis. PMN diagnosis was confirmed by light microscopy analysis of renal biopsies for all patients (inclusion criteria). Screening for secondary membranous nephritis was performed for all patients. All PMN patients were negative for antinuclear antibody (ANA), hepatitis B, C and HIV infections and had no evidence of a neoplasm (exclusion criteria). The clinical and biological features of PMN patients are recorded in Table 1. The treatment of PMN patients was based on the KDIGO guidelines [9]. The KDIGO criteria [9] were used to classify the disease outcome as follow:

*Complete remission*: defined as a 24-hour proteinuria < 0.3 g with normal serum albumin and serum creatinine concentrations.

*Partial remission*: defined as a 24-hour proteinuria < 3.5 g and urinary protein reduction by ≥ 50% with an improvement of serum albumin and a stable renal function (≤ 25% increase in serum creatinine.

*Non-remission* (non-response): defined as a 24-hour proteinuria > 3.5 g or persistence of ≥ 50% baseline proteinuria or > 25% increase in serum creatinine.

Forty-three (63.2%) PMN patients received initial therapy consisting of alternating monthly cycles of corticosteroids and cyclophosphamide (Table 1). Twelve (17.6%) patients had resistant PMN to initial therapy and were consecutively treated by cyclosporine or tacrolimus.

The SLE (n = 30) patients were retrospectively investigated in the nephrology and internal medicine department of Charles Nicolle Hospital in Tunis, and were diagnosed according to the Committee of the American College of Rheumatology criteria [10] and treated by an association of corticosteroids (prednisone) with antimalarial (hydroxychloroquine). All selected SLE patients had a class V membranous lupus nephritis as defined by the revisited SLE glomerulonephritis classification [11]. Mean age of the SLE patients was at 33.26 ± 13.75 years with a sex ratio (Men/Women) of 0.25 (6/24).

**Table 1. Clinical and biological characteristics of the PMN patients.**

| PMN patients | n = 68 |
|---|---|
| Sex-ratio (Male/Female) | 1.72 (43/25) |
| Onset age ± SD (years) | 42.9 ± 15.8 |
| Follow-up [1st– 3rd quartiles] (months) | 12 [10.25–16] |
| High blood pressure | 34 (50%) |
| Hematuria | 45 (66.17%) |
| Baseline Serum proteins ± SD (g/L) | 52.55 ± 8.71 |
| Albumin ± SD[a] (g/L) | 23.15 ± 7.67 |
| Baseline Proteinuria [1st– 3rd quartiles] (g/24h) | 4.17 [2.42–6] |
| Nephrotic syndrome at diagnosis | 42 (61.8) |
| Baseline Serum creatinine [1st– 3rd quartiles] (μmol/L) | 72.5 [62.25–100.45] |
| Baseline Creatinine clearance (CC) ± SD (ml/min/1.73 m$^2$) | 94.96 ± 34.45 |
| Kidney failure (KF) at diagnosis | 32 (47.1%) |
| Stage of KF[b] at diagnosis | |
| Stage I: CC $\geq$ 90 | 36 (52.9%) |
| Stage II: 60 $\leq$ CC[c] < 90 | 22 (32.4%) |
| Stage III: 30 $\leq$ CC < 60 | 7 (10.3%) |
| Stage IV: 15 $\leq$ CC < 30 | 2 (2.9%) |
| Stage V: CC < 15 | 1 (1.5%) |
| Fibrosis in kidney biopsy | 52 (76.5%) |
| Initial therapy: Corticosteroid/Cyclophosphamide | 43 (63.2%) |
| Cyclosporine or Tacrolimus | 12 (17.6%) |
| PMN outcome | |
| Complete remission/spontaneous remission | 26 (38.2%)/25 (36.8%) |
| Partial remission | 30 (44.1%) |
| Non-remission (non-response) | 12 (17.6%) |

[a]SD, standard deviation.

[b]CC, creatinine clearance.

[c]KF, kidney failure.

Twenty patients with biopsy-proven TIN were retrospectively in the nephrology and internal medicine department of Charles Nicolle Hospital in Tunis. Mean age of TIN patients was at 45.85 ± 12.82 years with a sex ratio (Men/Women) of 1.5 (12/8).

Controls (n = 30) were healthy subjects matched in age, gender and ethnicity with the PMN patients. Ethnicity (Tunisian) of both patients and controls was determined by an oral questionnaire. None of the healthy subjects had any evidence of personal or family history of autoimmune disease. Mean age of controls was at 29.96 ± 8.67 years with a sex ratio (Men/Women) of 1.14 (16/14).

## Ethics statement

All patients and controls gave written informed consent to participate in the study, and the local Ethics' committee of Charles Nicolle Hospital approved this study. No benefits in any form have been received or will be received from a commercial party related directly or indirectly to the subject of this manuscript.

## Methods

**Blood sampling.** Blood samples from all subjects were collected before starting immuno-suppressive therapy at the Immunology laboratory of the Charles Nicolle Hospital on EDTA tubes. Upon receipt, each tube was centrifuged at 3000 rpm for 15 minutes. The plasma, aliquoted in 1 ml tubes and frozen at -80°C, was used for the measurement of anti-PLA2R Ab. The cell pellet was used for the extraction of genomic DNA by the standard salting-out procedure [12].

**Anti-PLA2R antibody measurement.** The anti-PLA2R Ab assay was carried out by indirect ELISA using the commercial kit 'EA 1254–9601 G' (EUROIMMUN®, Medizinische Labordiagnostika AG, 23560 Lübeck, Germany) according to manufacturer instructions. At glance, the assayed samples were diluted at 1/100 in the sample diluent solution. Then, 100 μl of the 5 calibrators (2, 20, 100, 500 and 1500 RU/ml), positive and negative controls and all diluted samples were added to microplate wells containing recombinant PLA2R for a 30 minutes incubation. After 3 washes, 100μl of an anti-human-IgG/peroxidase conjugate solution were added for a second 30 minutes incubation. After 3 washing steps, 100μl of TMB substrate were added for a 15 minutes incubation followed by an addition of 100μl of an $H_2SO_4$ stop solution. Optical densities were read at 450 nm (reference: 620 nm) using a microplate photometer reader. As recommended by the manufacturer values $\geq$ 20RU/ml were considered positive.

**PLA2R rs4664308 (A>G +1624) SNP genotyping.** The PLA2R rs4664308 SNP (https://www.ncbi.nlm.nih.gov/snp/rs4664308) genotyping was carried out by a real-time PCR using the commercial kit 'Ref: 4351379' (Applied Biosystems® Inc, 500 Cummings Center, Beverly, MA 01915, USA). This kit is based on the 'TaqMan' technology which use two probes of reporter markers allowing the differentiation between homozygous and heterozygous samples. Amplification reactions were performed on an ABI 7500 real-time PCR system (Applied Biosystems® Inc) according to the manufacturer instructions. The results analysis was realized using the 'TaqMan® Genotyper Software'. This software allows allelic discrimination according to the type of acquisition: 1) VIC™ dye: allele n°1 or wild-type allele rs4664308*A and 2) FAM™ dye: allele n°2 or mutant allele rs4664308*G.

**Kidney biopsy and PLA2R mRNA quantification.** Kidney biopsy was performed for all PMN (n = 68) and TIN (n = 20) patients at the time of diagnosis. Total RNA was isolated from renal biopsies embedded in paraffin and fixed in formalin using the commercial kit 'RNeasy® FFPE Kit' (Qiagen Str. 1, 40724 Hilden, Germany) following the manufacturer instructions. At least 3 biopsy slices were used for RNA extraction for each patient. Reverse transcription for cDNA synthesis was made in one step together with the specific amplification of the target or the housekeeping genes. For cDNA synthesis we used the "Rotor-Gene SYBR Green RT-PCR master mix" (Qiagen) and random primers (48190–011, Invitrogen™). The PLA2R mRNA expression was evaluated by a real-time PCR based on SYBR Green technology on a Rotor Gene Q Thermocycler (Applied Biosystems, Inc). The relative expression of PLA2R has been estimated comparatively to a reference gene expressed in the podocytes, the podocin. The 'TaqMan' gene expression tests Hs00234853_m1 (PLA2R) and Hs00922492_m1 (Podocin) (Applied Biosystems, Darmstadt, Germany) were used. The mix used for reverse transcription and cDNA amplification was as follow: 1) Rotor-Gene SYBR Green RT-PCR Master Mix (2X): 12.5 μl, 2) Rotor-Gene RT-Mix: 0.25 μl, 3) Random primers (150 ng/μl): 1.25 μl, 4) Extracted RNA (concentration: 2 to 4 ng/μl): 5 μl, 5) Specific primers Hs00234853_m1 (PLA2R) or Hs00922492_m1 (Podocin): 1.25 μl and 6): RNase free water: 4.75 μl. Thermal cycling was performed with an initial reverse transcription step at 55°C for 10 minutes, an activation step at 95°C for 5 minutes and then 40 cycles of denaturation at 95°C for 5 seconds and annealing/

extension at 60˚C for 10 seconds. All samples were tested in triplicate. The Ct values of an internal control for both target gene (PLA2R) and reference gene (Podocin) were determined. The relative expression for the target gene comparatively to the reference gene was given by the following formulae: $2^{-\Delta\Delta Ct}$.

We used podocin as a housekeeping gene in order to replicate the study of Hoxha et al [8]. Comparison of the CT-values of podocin between TIN and PMN patients showed no significant difference ($p$ = 0.284). Moreover, we tested the correlation between CT-values of podocin and 24h-proteinuria, and there was no significant correlation, Spearman Rho = -0.112, $p$ = 0.362. Likewise, podocin CT-values were not associated to nephrotic syndrome at the moment of kidney biopsy, $p$ = 0.865. Since the CT-values of podocin did not vary between PMN and TIN patients and were not influenced by 24h-proteinuria or the presence of nephrotic syndrome we concluded that podocin was a good housekeeping gene in our study. Moreover, comparison of podocin and the RNA 18S Ct-values in 26 PMN patients showed a significant correlation; Spearman Rho = 0.731, $p$ = 2.2E-5.

## Statistical analysis

Statistical evaluation was carried out using the Statistical Package for the Social Sciences (SPSS) version 11 (IBM®, Armonk, USA). $p$-values <0.05 were considered significant.

Chi-square or Fisher exact tests were used to test the association between categorical variables. Odds ratio (OR) together with 95% confidence intervals [95% CI] were calculated to estimate the strength of the association.

ANOVA, Mann-Whitney U and Kruskal-Wallis tests were used to analyze quantitative variables as appropriate.

The Spearman's rank correlation coefficient was determined to test the association between quantitative variables.

Receiver-operating characteristic (ROC) curves were used to assess performances (specificity and sensitivity) of anti-PLA2R Ab and PLA2R mRNA tests in diagnosing PMN and detecting the non-remission outcome.

The Hardy–Weinberg equilibrium was analyzed using an online HWE software (https://ihg.helmholtz-muenchen.de/cgi-bin/hw/hwa1.pl).

## Results

As summarized in Table 1, mean onset-age of the PMN patients was at 42.9 ± 15.8 years with a sex ratio (Men/Women) of 1.72 (43/25). Median follow-up of the PMN patients after diagnosis was 12 [10.25–16] months. Clinically, 34 (50%) patients had high blood pressure and hematuria was found in 45 (66.17%) cases. Forty-two (61.8%) PMN patients had a nephrotic syndrome with a mean serum albumin of 23.15 ± 7.67 g/l and a median 24-hour proteinuria of 4.17 [2.42–6] g.

Besides, 32 (47.1%) patients had a kidney failure at the moment of PMN diagnosis with a median serum creatinine of 72.5 [62.25–100.45] μmol/l. A complete remission was obtained in 26 (38.2%) patients, while 30 (44.1%) patients had evolved into partial remission and 12 (17.6%) were non-responders (non-remission). PMN outcome was not associated to baseline 24-hour proteinuria, $p$ = 0.422.

### Anti-PLA2R antibody in PMN, SLE and controls

Anti-PLA2R Ab were positive in 54 (79.4%) PMN patients, while all SLE patients and controls were negative, $p$<0.0001. Moreover, anti-PLA2R Ab levels were significantly higher in PMN patients (134.85 [41.25–256.97] RU/ml) than in SLE patients (3.35 [2.3–4.35] RU/ml) and

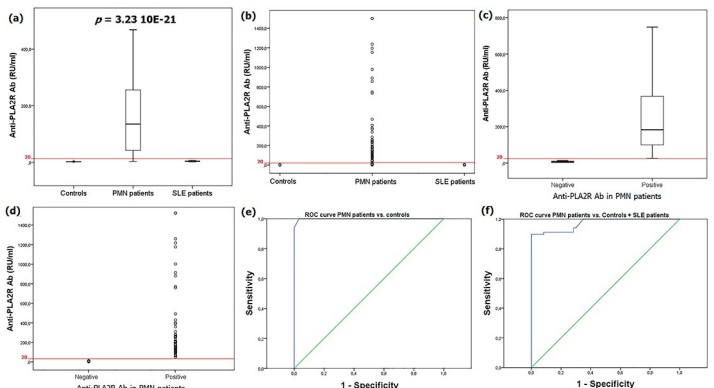

**Fig 1. Comparative analysis of Anti-PLA2R Ab in PMN, SLE and controls. (a)** Box plot underlining significant higher anti-PLA2R Ab levels in PMN patients. **(b)** Dot plot highlighting the anti-PLA2R antibody level in PMN, SLE and controls. **(c)** Box plot showing anti-PLA2R antibody level in positive and negative PMN patients. **(d)** Dot plot highlighting the anti-PLA2R antibody level in positive and negative PMN for anti-PLA2R Ab. **(e)** ROC curve for anti-PLA2R Ab (PMN vs. Controls): AUC = 99.19% [0.996–1] $p<0.0001$; Cut-off = 2.6 RU/ml, specificity = 100%, sensitivity = 94.1%. **(f)** ROC curve for anti-PLA2R Ab (PMN vs. SLE + Controls): AUC = 97.1% [0.947–0.995], $p<0.0001$; cut-off = 7.55 RU/ml, specificity = 100%, sensitivity = 89.7%.

controls (2 [2–2.3]), $p<0.0001$ (Fig 1). In order to determine the ability of anti-PLA2R in identifying PMN, a ROC curve was used with an area under curve (AUC) of 99.19% [0.996–1], $p<0.0001$ (Fig 1e). For a 100% specificity, the maximum of sensitivity was about 94.1% at a cut-off value of 2.6 RU/ml.

Considering SLE patients as controls a 2nd ROC curve was built with an AUC of 971% [0.947–0.995], $p<0.0001$ (Fig 1f). Given 100% specificity, the sensitivity was at 89.7% at a threshold of 7.55 RU/ml.

## Results of the anti-PLA2R antibody in PMN patients

The presence of anti-PLA2R Ab was significantly associated to a younger onset age (41 ± 14.95 vs. 50.4 ± 17.42 years), $p = 0.042$. Inversely, anti-PLA2R Ab were not associated to either nephrotic syndrome or kidney failure at diagnosis (Table 2).

Besides, anti-PLA2R Ab positivity was significantly associated to the non-remission outcome, $p = 0.002$ (Table 2). Moreover, non-remission patients had significantly higher levels of anti-PLA2R Ab (742.25 RU/ml) comparatively to patients with complete remission (41.65 RU/ml) and partial remission (183.2 RU/ml), $p<0.0001$ (Table 2) (Fig 2a and 2b). Multivariable analysis using the following covariables: baseline 24h-protéinuria, rs4664308 SNP, the presence of nephrotic syndrome at diagnosis, the presence of kidney failure at diagnosis confirmed the significant associations of anti-PLA2R Ab presence and level with PMN outcome. Then a ROC curve was performed to assess the ability of anti-PLA2R Ab levels at PMN diagnosis in detecting the non-remission outcome. With an AUC of 92.9% [0.869–0.989], $p<0.0001$, specificity and sensitivity were respectively 87.5% and 100% at a cut-off value of 239.95 RU/ml (Fig 2c). Contrariwise, anti-PLA2R Ab levels, and baseline 24h-proteinuria were not correlated, Spearman Rho = -0.080, $p = 0.515$.

## PLA2R rs4664308 SNP genotyping

The PLA2R rs4664308*A/A homozygous wild-type genotype was significantly more frequent in PMN patients (62%) than in controls (43%) and in SLE patients (40%); $p = 0.0018$ and $p = 0.009$, respectively (Table 3). Likewise, the frequency of the PLA2R rs4664308*A wild-type

**Table 2. Analysis of anti-PLA2R Ab with PMN patients features.**

| | Anti-PLA2R +n = 54 | Anti-PLA2R -n = 14 | p |
|---|---|---|---|
| **Gender** | | | |
| Male (n = 43) | 33 (61.1%) | 10 (71.4%) | 0.47 |
| Female (n = 25) | 21 (38.9%) | 4 (28.6%) | |
| **Onset age (years)** | 41 ± 14.95 | 50.4 ± 17.42 | **0.042** |
| **High blood pressure** | 26 (48.1%) | 8 (57.1%) | 0.54 |
| **Hematuria** | 36 (66.7%) | 9 (64.3%) | 0.867 |
| **Baseline Proteinuria (g/24h)** | 4.1 | 4.81 | 0.421 |
| **Nephrotic syndrome at diagnosis** | 32 (59.3%) | 10 (71.4%) | 0.4 |
| **Baseline Serum creatinine (μmol/L)** | 71.5 | 86 | 0.617 |
| **Baseline CC[a] ± SD (ml/min/1.73 m$^2$)** | 97.03 ± 33 | 86.99 ± 39.75 | 0.383 |
| **Kidney failure at diagnosis** | 23 (46.2%) | 9 (64.3%) | 0.147 |
| **PMN outcome** | | | |
| Complete remission | 15 (27.8%) | 11 (78.6%) | **0.002** |
| Partial remission | 27 (50%) | 3 (21.4%) | |
| Non-remission | 12 (22.2%) | 0 | |
| **PMN outcome** | Baseline anti-PLA2R Ab level (RU/ml) | | |
| Complete remission | 41.65 | | <**0.0001** |
| Partial remission | 183.2 | | |
| Non-remission | 742.25 | | |

[a]CC, creatinine clearance.

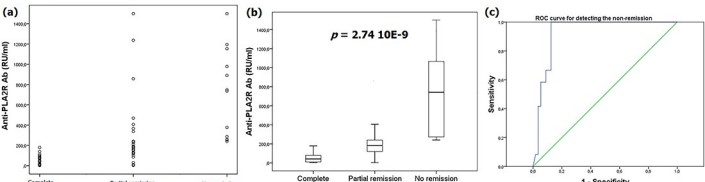

**Fig 2. Anti-PLA2R Ab and PMN outcome. (a)** Dot plot underlining anti-PLA2R antibody levels in terms of PMN outcome. **(b)** Box plot highlighting significant higher anti-PLA2R levels in non-remission patients. **(c)** ROC curve for anti-PLA2R Ab in predicting non-remission: AUC = 92.9% [0.869–0.989], p = 3.46 10E-6; Cut-off = 372.1 RU/ml, specificity = 91.1%, sensitivity = 66.7%.

**Table 3. PLA2R rs6448308 genotypes and alleles frequencies.**

| PLA2R Genotypes | Controls n = 30 | PMN n = 68 | SLE n = 30 | p[a] | OR [95% CI][a] | p[b] | OR [95% CI][b] |
|---|---|---|---|---|---|---|---|
| A/A | 13 (43%) | 42 (62%) | 12 (40%) | 0.0018 | – | 0.009 | – |
| A/G | 12 (40%) | 26 (38%) | 15 (50%) | | | | |
| G/G | 5 (17%) | 0 | 3 (10%) | | | | |
| **Alleles** | | | | | | | |
| A | 0.633 | 0.809 | 0.65 | 0.008 | 2.44 [1.24–4.82] | 0.016 | 2.27 [1.15–4.5] |
| G | 0.367 | 0.191 | 0.35 | | | | |

[a] PMN vs. Controls.

[b] PMN vs. SLE.

allele was significantly higher in PMN patients (0.809) than in controls (0.633) and SLE patients (0.65); $p = 0.008$, OR [95% CI] = 2.44 [1.24–4.82] and $p = 0.016$, OR [95% CI] = 2.27 [1.15–4.5], respectively (Table 3).

Analysis of the PLA2R rs4664308 SNP showed no association with any of the clinical and biological features of the PMN patients. Similarly, this SNP was not associated to PMN outcome in our patients.

## PLA2R mRNA levels in PMN and TIN

PLA2R mRNA relative levels were significantly higher in PMN patients (218.29 [66.05–486.07]) comparatively to TIN patients (22.09 [13.62–43.34]), $p<0.0001$ (Fig 3). So as to determine the performances of PLA2R mRNA quantification in diagnosing PMN, a ROC curve was built with an AUC of 90% [0.827–0.972], $p<0.0001$. For a 90% specificity, the sensitivity was about 80.9% at a threshold of 57.68 (Fig 3d).

PLA2R mRNA levels were significantly lower in patients with a kidney failure at diagnosis; 136.2 vs. 377.25, $p = 0027$ (Table 4). Also, PLA2R mRNA expression was significantly higher in patients with anti PLA2R ab; 331.36 vs. 37.92, $p<0.0001$ (Table 4) (Fig 3b). Moreover, PLA2R mRNA and anti-PLA2R Ab levels were significantly correlated to each other; Spearman Rho = 0.958, $p<0.0001$ (Fig 4). Inversely, PLA2R mRNA levels were not associated to the PLA2R rs4664308 SNP (Table 4). Inversely, PLA2R mRNA levels and baseline 24h-proteinuria were not correlated, Spearman Rho = -0.069, $p = 0.577$.

Non-remission patients had significantly higher levels of PLA2R mRNA (815.33) comparatively to patients with complete remission (66.55) and partial remission (331.36), $p<0.0001$ (Fig 5a and 5b). Multivariable analysis using baseline 24h-protéinuria, rs4664308 SNP, the presence of nephrotic syndrome at diagnosis and the presence of kidney failure at diagnosis as covariables confirmed the significant associations between PLA2R mRNA level and PMN

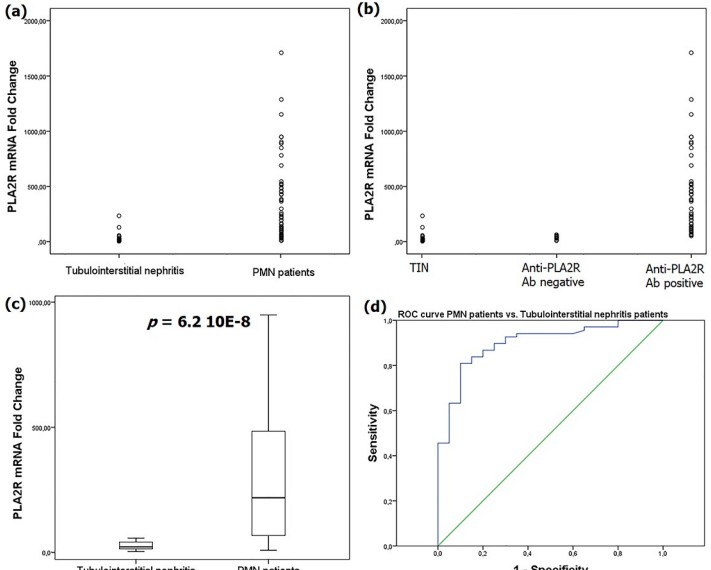

**Fig 3. PLA2R mRNA relative expression in PMN and TIN.** For each sample a relative expression of PLA2R to podocin was calculated. Then a fold-change for relative PLA2R expression was obtained from the calibrator. **(a)** Dot plot highlighting higher levels of renal PLA2R in PMN patients. **(b)** Dot plot showing PLA2R mRNA levels in TIN patients and PMN patients positive and negative for anti-PLA2R Ab. **(c)** Box plot emphasizing significant higher PLA2R mRNA levels in PMN patients. **(d)** ROC curve for PLA2R mRNA levels in PMN diagnosing: ROC curve: 90% [0.827–0.972], $p<0.0001$; cut-off = 57.68, specificity = 90%, sensitivity = 80.9%.

**Table 4. Analysis of renal PLA2R expression with PMN patients features.**

| | PLA2R mRNA level | *p* |
|---|---|---|
| **Male (n = 43)** | 192.67 | 0.258 |
| **Female (n = 25)** | 379.7 | |
| **High blood pressure + (n = 34)** | 226.75 | 0.783 |
| **High blood pressure–(n = 34)** | 194.01 | |
| **Hematuria + (n = 45)** | 225.97 | 0.683 |
| **Hematuria–(n = 23)** | 166.57 | |
| **Nephrotic syndrome at diagnosis + (n = 42)** | 210.66 | 0.86 |
| **Nephrotic syndrome at diagnosis–(n = 26)** | 218.29 | |
| **Kidney failure at diagnosis + (n = 32)** | 136.23 | **0.027** |
| **Kidney failure at diagnosis–(n = 36)** | 377.25 | |
| **PMN outcome** | | |
| Complete remission (n = 26) | 66.55 | **<0.0001** |
| Partial remission (n = 30) | 272.72 | |
| Non-remission (n = 12) | 815.33 | |
| **Anti-PLA2R Ab + (n = 54)** | 331.36 | **<0.0001** |
| **Anti-PLA2R Ab–(n = 14)** | 37.92 | |
| **rs4664306*A/A (n = 42)** | 136.23 | 0.472 |
| **rs4664308*A/G (n = 26)** | 246.42 | |

outcome. Thus, a ROC curve was built to evaluate the ability of PLA2R mRNA levels to detect the non-remission outcome (Fig 5c). With an AUC of 90% [0.827–0.973], $p<0.0001$, specificity and sensitivity were respectively 91.1% and 58.3% at a cut-off value of 535.67.

## Discussion

PMN is an antibody-mediated glomerular disease caused by the formation of a large number of IgG4 immune-complexes in the subepithelial space of the glomerular basement membrane (GBM) [3]. In the majority of PMN patients, these IgG4 colocalize with the PLA2R [3] which have been reported to be overexpressed on podocytes [4]. In addition to their diagnostic specificity for PMN, anti-PLA2R Ab have also been reported to be of a prognostic significance and predicted a non-remission outcome [13].

Besides, and like other autoimmune disease, the etiology of PMN is complex including a multigenetic background and environmental factors [7]. Among genetic factors, HLA and PLA2R genes were the most associated with PMN risk. Indeed, in a study including 3 European cohorts [6], Stanescu et al noted that simultaneous presence of HLA-DQA1 rs218668*A/A and PLA2R rs4664308*A/A genotypes increased the risk of PMN by 78.46-fold. This peculiar finding suggests that an overexpression and/or increased immunogenicity of PLA2R combined with a higher ability of its presentation to T cells conferred by HLA-DQA1 risk allele might be responsible for PMN occurrence.

In this study, 68 PMN patients were enrolled. Only 42 (61.8%) patients were nephrotic at the moment of diagnosis. Nevertheless, only 12 (17.6%) PMN patients evolved to a non-remission. Of note, PMN outcome was not influenced by baseline 24-hour proteinuria, $p = 0.422$. This is an uncommon finding as proteinuria is generally recognized as a major prognostic factor [14]. This issue could be due to the generally good prognosis of our patients and the small size of our cohort.

In the present study, 79.4% of PMN patients were positive for anti-PLA2R Ab. This anti-PLA2R Ab frequency in our PMN patients corroborates those of previous reports in which it

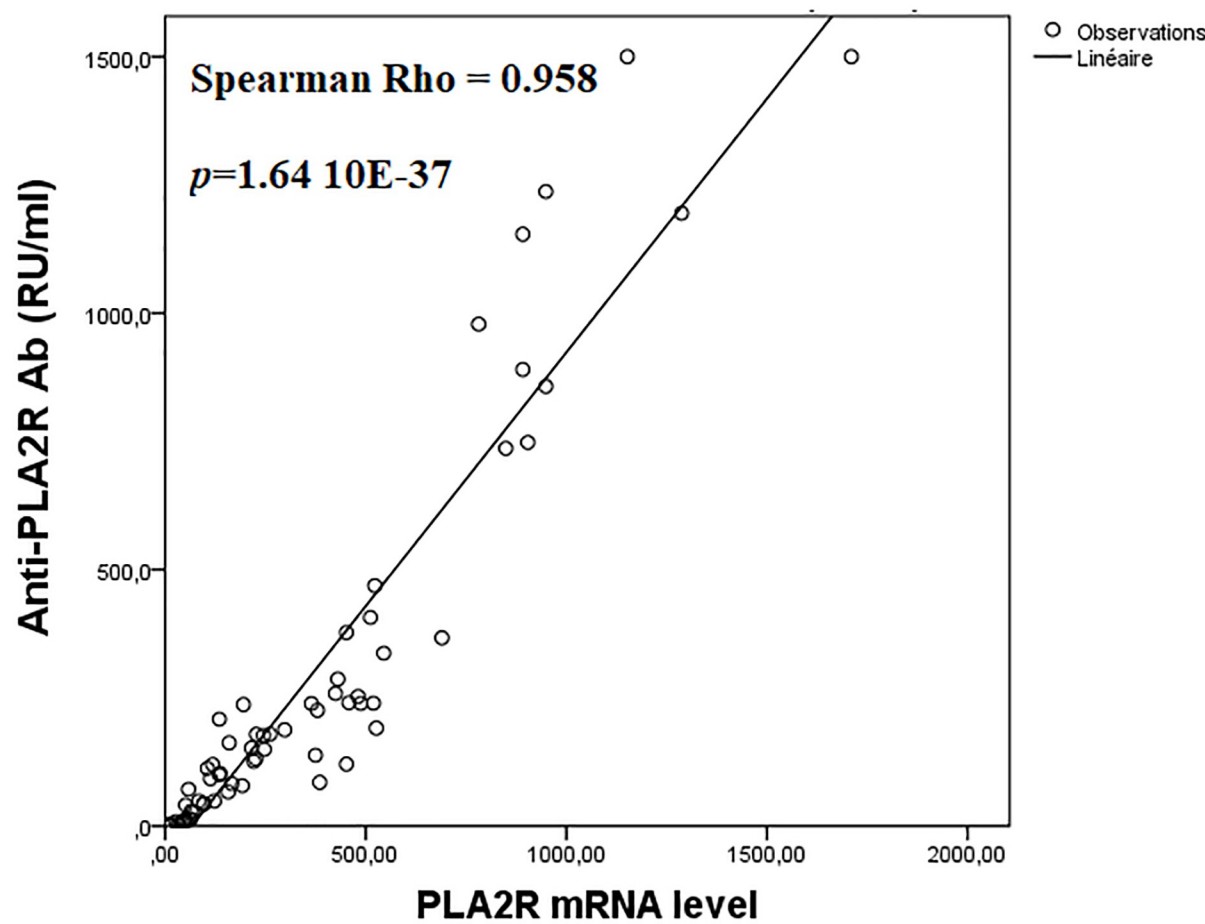

**Fig 4. Significant correlation between PLA2R mRNA and anti-PLA2R Ab levels.** All PMN patients (positive and negative for anti-PLA2R Ab) were plotted for correlation with renal PLA2R level.

varied between 52% and 86% [7]. Also, none of the SLE patients and the control subjects had anti-PLA2R Ab in their sera. Of note some previous studies reported a rare presence of anti-PLA2R Ab in SMN patients. In fact, Garcia-Vives et al [15], Radice et al [16], Qin et al [17] and Hofstra et al [18] noted that SLE patients with membranous lupus nephritis (MLN) had anti-PLA2R Ab at frequencies of 5.3%, 5.4%, 5% and 5%, respectively. This rare positivity of anti-PLA2R Ab during MLN could be due to a bystander activation of self-reactive T and B cells. Indeed, the *in-situ* deposition of DNA/anti-DNA-Ab immune complexes in the MBG

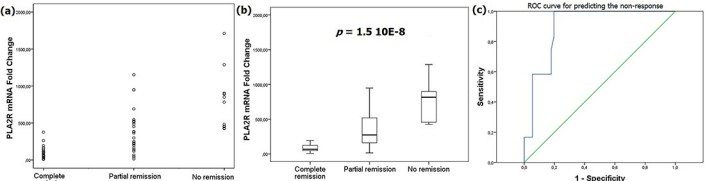

**Fig 5. PLA2R mRNA relative expression and PMN outcome. (a)** Dot plot showing the renal PLA2R expression in terms of PMN outcome. **(b)** Box plot highlighting significant higher PLA2R mRNA levels in non-remission patients. **(c)** ROC curve for PLA2R mRNA levels in predicting non-remission: AUC = 90% [0.826–0.973], $p < 0.0001$; cut-off = 535.67, Sensitivity = 58.3%, specificity = 91.1%.

could modify the podocyte biology and cause conformational and/or quantitative changes of PLA2R expression and therefore could trigger the synthesis of anti-PLA2R Ab. Some studies reported, although seldom, the presence of anti-PLA2R Ab in patient with SMN associated with hepatitis B [16], and neoplasms [16,17]. Of note, some authors suggested that the presence of anti-PLA2R antibodies in patients with SMN might result from the co-incidental simultaneous development of PMN and a systemic disease, such as SLE or a malignancy [18]. In fact, Qin et al [17], showed that in anti-PLA2R positive patients, proteinuria persisted despite tumor resection.

In this study, PMN patients had significantly higher levels of anti-PLA2R Ab comparatively to controls and SLE patients, $p<0.0001$. Subsequently, a ROC curve was used and revealed for 100% specificity a sensitivity of 94.1% at a cut-off value of 2.6 RU/ml. Nevertheless, the sensitivity at the cut-off value of 20 RU/ml was 79.4%. These performances agree with those estimated in a recent meta-analysis of 28 published studies [19] in which the estimated pooled sensitivity and specificity were 65% [0.53–0.67] and 97% [0.97–0.98], respectively. Of note, to obtain this good sensitivity of 94.1% the threshold was set at 2.6 RU which is lower than that of the manufacturer's recommendations (20 RU/ml). The definition of a new cut-off has also been recommended by the authors of 2 published studies [20,21]. In fact, in a study carried out on 67 Italian PMN patients and 36 controls, the sensitivity was 88.1% with a specificity of 96% at a threshold value of 2.7 RU/ml [20]. Similarly, in the study of Liu et al [21] performed in 119 Chinese PMN patients and 22 controls, the sensitivity and the specificity were 78.9% and 91.7%, respectively, at a cut-off value of 2.6 RU/ml. Besides, in the present study when we considered the 30 SLE patients, the sensitivity and specificity were 89.7% and 100% at a cut-off value of 7.55 RU/ml. Thus, if we set the threshold of anti-PLA2R Ab positivity at 2.6 RU/ml, 60% (18/30) of our SLE patients would be considered positive. Hence, the interval from 2.6 to 7.55 RU/ml could be considered as a gray area in which the result would be interpreted as equivocal. Nevertheless, the systematic screening for ANA as well as the clinical and biological features can easily differentiate PMN from SLE with MLN.

In the present study, the presence of anti-PLA2R was associated to an earlier onset (41 ± 14.95 vs. 50.4 ± 17.42 years), $p = 0.042$. This peculiar finding has not been noted in previous reports [22] and needs to be confirmed in larger independent cohorts. Also, the presence of anti-PLA2R Ab at diagnosis was significantly associated with the non-remission outcome, $p = 0.002$. Even there is a limitation due to the presence of some non-nephrotic patients at baseline, our result corroborates those of previous studies which were confirmed by 2 recent meta-analyzes [23,24]. In fact, in the metanalysis of Liang et al [23] including 11 published studies, the presence of anti-PLA2R Ab decreased the chance of clinical remission by 0.76-fold [0.68–0.86], $p<0.0001$. Similarly, the meta-analysis of Li et al [24] showed that PMN patients with negative baseline anti-PLA2R Ab had 1.65 times (95% CI: 1.46–1.87, $p<0.05$) increased chance of reaching complete remission and 1.93 times (95% CI: 1.53–2.45, $p<0.05$) increased chance of achieving spontaneous remission. Moreover, in the present study, anti-PLA2R Ab levels were significantly higher in non-remission patients, $p<0.0001$. Thus, for a threshold of 239.95 RU/ml, the sensitivity and the specificity of anti-PLA2R Ab in predicting non-remission were 100% and 87.5%. Significant higher levels of anti-PLA2 Ab in non-remission patients was also reported by previous studies [25,26] and confirmed by the meta-analysis of Liang et al [23]. In this meta-analysis, PMN patients with high titers of PLA2R Ab had 0.72-fold (95% CI: 0.59–0.87, $p = 0.0006$) decreased chance of achieving clinical remission [23]. In a recent study [26], sera from patients with high levels of anti-PLA2R Ab showed a multiple reactivity to CTLD1 and CTLD7 domains in addition to CysR domain, $p<0.001$. These so-called *spreaders* patients had more severe proteinuria and epitope-spreading was an independent risk factor for decreased remission rate; OR [95% CI] = 0.16 [0.04–0.72], $p = 0.02$

[27]. The authors suggested that epitope-spreading might be relevant in patients with anti-PLA2R Ab levels ≥ 50 RU/ml. In addition, Ramachandran et al [28] and Segarra-Medrano et al [29] reported temporal associations between the reduction of anti-PLA2R Ab levels and the achievement of clinical remission. Thus, an 'immunological remission' is a relevant therapeutic goal in anti-PLA2R Ab positive patients.

In this study, the PLA2R rs4664308*A wild-type allele was significantly associated to PMN risk in our patients, $p = 0.008$, OR [95% CI] = 2.44 [1.24–4.82]. The protective role of the rs4664308*G mutant allele was also reported by several previous studies [6,30–33]. In fact, the OR of rs4664308*A wild-type allele for PMN varied between 1.17 and 2.76 [6,30–33]. Inversely, in our patients, the PLA2R rs4664308 was not associated either to 24-h proteinuria rate or serum creatinine levels. Moreover, this rs4664308 SNP did not influence the disease outcome in our patients which corroborates previous reports [30–32]. Besides, the PLA2R SNP rs4664308 was neither correlated to the presence of anti-PLA2R Ab nor to their levels in our patients. Inversely, and in only one study [30] performed in 71 Chinese PMN patients, the high-risk rs4664308*A/A genotype was associated to significant increased frequency of anti-PLA2R Ab while the other studies did not show a such association [31,32]. Of note, if the PLA2R rs4664308 SNP was associated to PMN risk in all studies, its functional influence on PLA2R expression is unknown. As mentioned above this intronic rs4664308 (+1624 A>G) SNP is correlated with the rs3749117 SNP (r2 = 0.70) which encodes a nonsynonymous amino acid substitution (M292V) in the CTLD1 domain [7] which could decrease its immunogenicity and may explain in part the rs4664308*G allele protective effect. Indeed, several published studies [6,30,33,34] reported a significant increased PMN risk conferred by the PLA2R rs3749117*T wild-type allele. However, it remains to determine how this nonsynonymous mutation influence the PLA2R expression and decrease the PMN susceptibility.

In the present study, the relative PLA2R expression in kidney was significantly increased in PMN patients, $p<0.0001$. Subsequently, a ROC curve was built and revealed for 90% specificity a sensitivity of 80.9% for PMN diagnosing at a cut-off value of 55.68. To our knowledge this is the first study to report significant increased renal PLA2R mRNA levels in PMN patients. Of note, this finding agrees with the enhanced PLA2R staining in biopsies from PMN patients reported in previous studies using immunohistochemistry (IHC) and/or direct immunofluorescence (DIF) [8,35–39]. Inversely, Hoxha et al [8] found no correlation between PLA2R staining by IHC and PLA2R mRNA levels. This issue could be due to small numbers of compared groups (only 6 in both groups) [8]. Moreover, in the study of Hoxha et al [8], the compared groups included only MN patients while in our study the controls were TIN patients which could explain the discrepancy between the results. The investigators noted that their finding does not exclude the possibility of increased post-transcriptional synthesis or any other mechanism that may increase the availability of PLA2R in the podocyte membrane [8]. Furthermore, we observed a significant inverse association between PLA2R mRNA levels and the presence of baseline renal failure; 34.77 vs. 76.99, $p = 0.027$. The decrease in PLA2R expression could be explained by the destruction of podocytes concomitant with the onset of renal failure. As the renal PLA2R expression is significantly correlated to anti-PLA2R Ab levels, and since this autoantibody level is associated with the non-remission outcome, this apparently paradoxical finding can be explained by the fact that lower mRNA levels induce less anti-PLA2R Ab and could be predictive of a better outcome despite its association with a baseline renal failure. However, this association was not reported in previous PLA2R-expression studies by IHC [8,38] or DIF [39]. The studied PLA2R rs4664308 SNP did not influence the renal PLA2R expression in our PMN patients. This finding indicates that the rs4664308*G protective allele might rather decrease the PLA2R immunogenicity than its expression.

In this study, the presence of PLA2R Ab was associated to higher levels of PLA2R mRNA; 331.36 vs. 37.92, $p < 0.0001$. Accordingly, PLA2R mRNA levels were significantly correlated to anti-PLA2R Ab titers; Spearman Rho = 0.958, $p < 0.0001$. These significant associations between PLA2R mRNA and anti-PLA2R Ab suggest that an increase in renal PLA2R transcription could trigger in part anti-PLA2R Ab production. Likewise, previous reports showed high concordance rates between enhanced PLA2R staining and the presence of anti-PLA2R Ab [8,35,36].

In the present study, higher levels of PLA2R mRNA were significantly associated to non-remission outcome, $p < 0.0001$. Then a ROC curve showed at a cut-off value of 535.67 a sensitivity of 58.3% for a specificity of 91.1% in predicting non-remission. Nevertheless, the presence of some non-nephrotic patients at baseline remains a limitation in this regard. The relationship between renal PLA2R expression and PMN outcome have been disparately estimated. In a study including 108 French MN patients [37] the intensity of PLA2R staining by DIF was similar in remission and non-remission patients. The study of Liu et al [39] in 122 Chinese PMN patients in which renal PLA2R expression was assessed by DIF showed that the remission rate in PLA2R-positive patients was significantly higher; 89.9% vs. 54.4%, $p < 0.05$. Inversely, Xu et al [40] reported a significant lower rate of complete remission in patients with positive PLA2R staining by IHC; 3.2% vs. 42.9%, $p = 0.004$. These conflicting results need replications in independent cohorts with a simultaneous PLA2R mRNA quantification.

## Conclusions

Based on these findings, the PLA2R rs6446308 SNP is associated with PMN susceptibility in Tunisians. Anti-PLA2R Ab and renal PLA2R mRNA could be useful markers for PMN outcome predicting.

## Supporting information

**S1 File. Patient consent form.** This document describes the consent that was signed by each patient included in this study.
(PDF)

**S2 File. Database of PMN patients, SLE patients and controls.** This file depicts the data acquired from the cohorts of patients with primary membranous nephritis, patients with systemic lupus erythematosus and controls.
(XLS)

**S3 File. Database of controls.** This file shows the results of PLA2R mRNA levels in biopsies from patients with primary membranous nephritis and tubulointerstitial nephritis patients.
(XLS)

## Author Contributions

**Conceptualization:** Tarak Dhaouadi, Raja Trabelsi, Hanene Gaied, Imen Sfar, Rym Goucha, Fethi Ben Hamida, Taieb Ben Abdallah, Yousr Gorgi.

**Data curation:** Jihen Abdellatif, Raja Trabelsi, Hanene Gaied, Sameh Chamkhi, Rym Goucha, Fethi Ben Hamida.

**Formal analysis:** Tarak Dhaouadi, Sameh Chamkhi, Imen Sfar.

**Investigation:** Tarak Dhaouadi, Jihen Abdellatif, Sameh Chamkhi, Imen Sfar, Yousr Gorgi.

**Methodology:** Tarak Dhaouadi, Imen Sfar.

**Project administration:** Imen Sfar, Yousr Gorgi.

**Software:** Tarak Dhaouadi.

**Supervision:** Yousr Gorgi.

**Validation:** Tarak Dhaouadi, Imen Sfar, Yousr Gorgi.

**Visualization:** Imen Sfar, Yousr Gorgi.

**Writing – original draft:** Tarak Dhaouadi.

**Writing – review & editing:** Tarak Dhaouadi.

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
