## [Decision Letter · Decision Letter 0]

15 Jul 2020

PONE-D-20-17580

PLA2R antibody, PLA2R rs4664308 polymorphism and PLA2R mRNA levels in Tunisian patients with primary membranous nephritis

PLOS ONE

Dear Dr. Dhaouadi,

Thank you for submitting your manuscript to PLOS ONE. After careful consideration, we feel that it has merit but does not fully meet PLOS ONE’s publication criteria as it currently stands. Therefore, we invite you to submit a revised version of the manuscript that addresses the points raised during the review process.

**The manuscript focuses on a topic of current interest. However, the paper has some important limitations and methodological flaws that should be addressed to improve the quality of the study and support sound conclusions. To mention few of them, i) need to modify the methods and results section of the Abstract; ii) unclear why in the Introduction the authors want to link Treg regulation with PLA2R expression; ii) concern about the suggestion by the authors that both the autoimmune phase and the effector pathological phase of autoantibodies acting on podocytes occur within the glomeruli; iv) concern about the fact that they acknowledge that 62% of primary membranous nephropathy patients were nephrotic at baseline, which introduces a major bias when looking at association between biomarkers and clinical outcomes; v) unclear how many patients were treated with the uncommon combination of prednisone and hydroxychloroquine; vi) unclear why the follow-up time for patients is quite short (median of 12 months); vii) unclear why the SLE patients were used as controls for ELISA assays, while TIN patients were exclusively used for mRNA analysis; vii) unclear why they have analyzed the SNP rs4664308 but not the other one in linkage disequilibrium rs3749117; ix) need to demonstrate that the authors’ new findings are relevant to primary membranous nephropathy disease and correlate to antibody level and outcome; x) unclear why podocin was used as a “reference” gene; xi) need to validate their findings using accurate housekeeping genes in their kidney biopsy samples; xii) need to provide the correlation between PLA2RmRNA and proteinuria and antibody level and proteinuria; xiii) need to analyze the correlation between proteinuria at baseline and clinical outcome, and demonstrate whether anti-PLA2R titer is still predictive; xiv) unclear why the authors chose a group with tubulointerstitial nephritis for comparison when evaluating the mRNA expression of PLA2R; xv) unclear whether the lupus patients are pure class V; xvi) unclear whether in patients that were considered PLA2R negative in the primary membranous nephropathy, the kidney biopsy stain negative of PLA2R; xvii) need to look for a cut off that is more sensitive than specific for PLA2R antibody.**

We look forward to receiving your revised manuscript.

Kind regards,

Giuseppe Remuzzi

Academic Editor

PLOS ONE

Journal Requirements:

2. We note that you are reporting an analysis of a microarray, next-generation sequencing, or deep sequencing data set. PLOS requires that authors comply with field-specific standards for preparation, recording, and deposition of data in repositories appropriate to their field. Please upload these data to a stable, public repository (such as ArrayExpress, Gene Expression Omnibus (GEO), DNA Data Bank of Japan (DDBJ), NCBI GenBank, NCBI Sequence Read Archive, or EMBL Nucleotide Sequence Database (ENA)). In your revised cover letter, please provide the relevant accession numbers that may be used to access these data. For a full list of recommended repositories, see http://journals.plos.org/plosone/s/data-availability#loc-omics or http://journals.plos.org/plosone/s/data-availability#loc-sequencing.

3. In your Methods section, please provide additional information about the participant recruitment method and the demographic details of the participants with PMN and SLE, and the healthy volunteers. Please ensure you have provided sufficient details to replicate the analyses such as: a) the recruitment date range (month and year), b) a description of any inclusion/exclusion criteria that were applied to participant recruitment, and c) a description of how participants were recruited.

"This study was granted and supported by the Research Laboratory in Immunology of Renal

Transplantation and Immunopathology (LR03SP01), and the Research Laboratory of Kidney

Diseases (LR00SP01), Charles Nicolle Hospital, Tunis El Manar University, Tunisia."

5. Your ethics statement must appear in the Methods section of your manuscript. If your ethics statement is written in any section besides the Methods, please move it to the Methods section and delete it from any other section. Please also ensure that your ethics statement is included in your manuscript, as the ethics section of your online submission will not be published alongside your manuscript.

Reviewers' comments:

Reviewer's Responses to Questions

**Comments to the Author**

1. Is the manuscript technically sound, and do the data support the conclusions?

Reviewer #1: Partly

Reviewer #2: Partly

2. Has the statistical analysis been performed appropriately and rigorously? 

Reviewer #1: Yes

Reviewer #2: Yes

3. Have the authors made all data underlying the findings in their manuscript fully available?

Reviewer #1: Yes

Reviewer #2: Yes

4. Is the manuscript presented in an intelligible fashion and written in standard English?

Reviewer #1: Yes

Reviewer #2: No

5. Review Comments to the Author

Reviewer #1: This manuscript by Dhaouadi et al is a single center study of a cohort of patients with primary membranous nephropathy in which the authors have analyzed the role of anti-PLA2R detection and titer, of one relevant PLA2R SNP and the level of PLA2R mRNA in kidney biopsy to evaluate the role of these parameters as possible biomarkers of susceptibility, disease severity and clinical outcome in this Tunisian population of patients. Overall, the study is well performed with standardized methods providing sound data. However, there are some limitations and methodological flaws that should be taken into account and addressed to improve the quality of the study.

Major comments:

1. Abstract: the methods section should be entirely modified as there is basically no information on the techniques used: ELISA, SNP genotyping, qPCR, etc. For instance, the last sentence says that TIN patients were used as controls but do not say that the PMN patients were analyzed by qPCR for PLA2R mRNA expression. Sounds odd. The result section should indicate how many patients were treated (or not). In the sentence "Moreover, PLA2R mRNA levels were significantly increased in non-remission patients; p=1.56 10E-8." "increased" should be replaced by "higher" and the fold factor may be indicated.

2. Introduction: In many occurrences, the only references cited are reviews. This is understandable, but this should be balanced by citing the original work when most appropriate. Some sentences contain information which are not linked, at least at first sight. For instance, in the sentence in lines 66-68 page 3, it is unclear why the authors wants to link Treg regulation with PLA2R expression?

3. In introduction and later in the discussion, the authors seem to suggest that BOTH the autoimmune phase and the effector pathological phase of autoantibodies acting on on podocytes occur within the glomeruli (introduction lines 82-86 page 4 and discussion lines 408-410 page 20). This is not documented and is in the opinion of this reviewer an over-simplification of the natural course of the disease. It is not because PMN is referred to an organ-specific disease, ie affecting the kidney, that the autoimmune response and the production of autoantibodies also occur in the kidney and more specifically the glomeruli or the podocyte surface. The autoimmune response may well develop elsewhere in the patient's body, even years before the clinical signs of the disease leading to podocyte injury and proteinuria, etc. Please clarify this point.

4. In introduction again, the authors should refer to the previous study of Hoxha et al (reference 32) already in the introduction, especially as they use the same technique to measure PLA2R mRNA expression.

5. Patients: The authors acknowledge that only 62% of PMN patients were nephrotic at baseline. This is OK but it is introducing a major bias when looking at association between biomarkers and clinical outcome, as the likelihood of spontaneous remission is known to be high for non-nephrotic patients. Accordingly, there is likely no need for treatment with immunosuppressants. About treatment, it is unclear how many patients were treated with the apparently very uncommon combination of prednisone and hydroxychloroquine. Is this treatment unique to this nephrology center? Please explain in details, provide regimen and references if any, etc. In table 1, how many patients enter into remission or not, depending on this treatment or due to spontaneous remission? The follow-up time for patients seems quite short (median of 12 months). Please explain why. Finally, it is unclear why the SLE patients were used as controls for ELISA assays while TIN patients were exclusively used for mRNA analysis. Why not use both categories of patients for both methods?

6. It is unclear why the authors have analyzed the SNP rs4664308 but not the other one in linkage disequilibrium rs3749117?

7. The most interesting and original part of the study is the analysis of PLA2R1 mRNA expression in biopsies from PMN versus TIN patients and the correlations observed between antibody titer and mRNA level and clinical outcome. However, the data appear controversial with those previously published by the German group (ref 32). It is thus critical to demonstrate that the authors' new findings are relevant to PMN disease and correlate to antibody level and outcome. In fact, the data may be due to some bias in the qPCR analysis. First, please give more details on the preparation of total mRNA (from how many biopsy slices were prepared total RNA), on cDNA synthesis (random primers, oligodT primers, amount of template used in qPCR analysis, etc). Second and most importantly, please explain why podocin was used as a "reference" gene? The authors should demonstrate that podocin is indeed a good housekeeping gene in their setting. There is literature showing that the expression of podocin varies in various types of glomerulonephritides including PMN and also inversely as a function of proteinuria. The expression level of podocin (and PLA2R1) may also vary from one biopsy to another depending on the number of glomeruli/kidney slice, etc. It might be in fact possible that the expression level of podocin inversely varies with that of PLA2R1 and this introduces some bias in the analysis. The authors should thus clearly validate their findings using accurate housekeeping genes in their kidney biopsy samples using for instance the Genorm kit (http://www.primerdesign.co.uk/products/9460-genorm-kits) and follow MIQE recommendations (https://pubmed.ncbi.nlm.nih.gov/19246619/). At the very least, they should use other housekeeping genes such as GAPDH or RNA 18S (see for instance PMID: 14569107 DOI: 10.1097/01.asn.0000090745.85482.06) and demonstrate that the expression level of podocin does not vary while the expression of PLA2R1 indeed varies between PMN patients versus controls (TIN and SLE if possible). Finally, it would be interesting to show that the level of mRNA of PLA2R1 is associated with the levels of the PLA2R1 protein by performing immunostaining of kidney biopsies using the standardized procedure used in clinical practice (Larsen et al, PMID: 23196797) or western blot analysis (ref 3) .

8. Data in figures 1, 2 , 3 and 4 are represented as box plots, but it would be more useful to show them as dot plots to visualize each patient. Furthermore, figure 1 should show the threshold value for anti-PLA2R1 used and show separately the PMN patients considered as anti-PLA2R1-positive versus negative. Figure 3 should also indicate the level of PLA2R mRNA for anti-PLA2R1 positive versus negative patients versus TIN patients, ie as three separate groups. This will show whether the anti-PLA2R1 positivity is associated with increased levels of mRNA, in addition to figure 4. In figure 4, were all PMN patients plotted for the correlation, or only anti-PLA2R1 positive patients?

9. Figure 4 shows an interesting correlation between PLA2R mRNA expression and antibody levels. What about the correlation between PLA2R mRNA and proteinuria? antibody level and proteinuria? Podocin level and proteinuria (using an appropriate housekeeping gene for normalization)?

10. Tables 1 and 2: percentages for males versus females better compare vertically. Parameters measured at baseline or diagnosis should be indicated as such. Probably true for many in tables 1 and 2.

11. The authors should analyze the correlation between proteinuria at baseline and clinical outcome, and then demonstrate whether anti-PLA2R1 titer is still predictive (multivariable analysis).

12. Discussion lines 340-342: There are former studies showing that age of patients anti-PLA2R negative versus positive are similar (PMID: 26142398 DOI: 10.1093/ndt/gfv228).

Minor comments:

1. "wild allele" should likely be "wild-type allele".

2. "Analytic" results sounds weird.

3. Figure 3 panel A, Y axis: is it really the fold-change that is measured or the relative expression of PLA2R1 to podocin? The legend and information in the figure panel should be made more explicite.

4. Please refrain from using "besides". Too many occurrences.

5. Figure 4: the legend in the Y axis is not correctly located.

Reviewer #2: The study by Dhaouadi et al. assesses the relationship between PLA2R antibody, rs4664306 SNP and PLA2R mRNA level with outcome. The only novel finding is the association between PLA2R mRNA outcomes. The other findings have been shown previously.

Major:

- When evaluating the mRNA expression of PLA2R, why did the authors chose a group with TIN for comparison? The comparative group should ideally be SLE patients with class V lupus who have a similar lesion but are PLA2R negative. The other comparative group would ideally be a negative control group which would be “normal” kidney tissue (e.g. from living donors, or noncancerous part of a nephrectomy sample). Choosing patients with TIN as a comparator does not make sense to me.

- Are the lupus patients pure class V? If so this needs to be specified. As it reads currently this is vague.

- In patients that were considered PLA2R negative in the primary MN group, did their kidney biopsy stain negative for PLA2R? If so this should be explicitly mentioned. If they were not stained this should be noted. This is important to know whether they were truly PLA2R negative or not.

- In Table 2, does the PLA2R antibody level for PMN outcome refer to PLA2R antibody at time of biopsy or at follow up? If at time of biopsy or baseline should specify.

- When evaluating for a cut off for PLA2R antibody in association with outcome a higher sensitivity (rather than specificity) would be more helpful for a practicing clinician. Let’s say if at 90% sensitivity the cut off for PLA2R antibody is 100 RU/ml, then one can feel fairly confident that if the patient has a PLA2R below that value the outcome will likely be good. I suggest looking for a cut off that is more sensitive than specific.

- How do authors explain that higher degree of CKD is associated with lower mRNA PLA2R yet patient with lower PLA2R mRNA expression are more likely to go into remission? This finding seems counterintuitive.

- The discussion from lines 307-319 does not seem to be relevant to this paper.

- Line 364 “autoantibody remission” is incorrect. The term is known as “immunological remission” and it is well established that immunological remission is important to achieve in patients with MN.

Minor:

- There are numerous grammatical errors that need to be adjusted, e.g. “had revolutionized” should be “has revolutionized”, “In fact, if several reports”, “if” should be deleted, “majority of patients carried serum PLA2R”, “carried” is an incorrect use of word and needs to be changed, etc…

- The way p values are reported is hard to understand. If p is very small should say p<0.0001.

- Line 81: need to clarify rs4664308 is *G and not *A, otherwise the text is confusing.

- The stage of CKD should be reported based on the CKD classifications of stage I-V and not as KF mild, moderate, etc…

6. PLOS authors have the option to publish the peer review history of their article (what does this mean?). If published, this will include your full peer review and any attached files.

Reviewer #1: No

Reviewer #2: No

---

## [Author Response · Author response to Decision Letter 0]

24 Aug 2020

Journal Requirements

1. We ensured that the manuscript meets PLOS ONE’s style requirements.

2. We are reporting three types of standard analyzes: a) an ELISA for anti-PLA2R antibody measurement, b) a real-time PCR for PLA2R rs4664308 polymorphism genotyping and c) For the renal PLA2R expression we used a commercial kit for RNA extraction, then we performed a real-time PCR using the TaqMan gene expression tests Hs00234853_m1 (PLA2R) and Hs00922492_m1 (Podocin), which were previously used in the study of Hoxha E, et al. Enhanced expression of the M-type phospholipase A2 receptor in glomeruli correlates with serum receptor antibodies in primary membranous nephropathy. Kidney Int 2012; 82(7): 797-804 doi: 10.1038/ki.2012.209. We added in the methods section the link to the rs4664308 SNP sequence data (https://www.ncbi.nlm.nih.gov/snp/rs4664308).

3. We provided the Methods section with the recruitment date range, inclusion and exclusion criteria for PMN patients and demographic details for all participants.

4. Funding information was deleted from the manuscript and was updated online. We confirm that the following Financial Disclosure statement is accurate:

This study was supported by the Research Laboratory in Immunology of Renal Transplantation and Immunopathology (LR03SP01), and the Research Laboratory of Kidney Diseases (LR00SP01), Charles Nicolle Hospital, Tunis El Manar University, Tunisia.

5. The ethics statement is in the subjects’ section within the Material and Methods chapter.

Comments to the Author

Review comments to the author

Reviewer #1:

Major comments

Comment 1: We provided the abstract with information on techniques used and we specified that PMN patients were analyzed for renal PLA2R expression. “increased” was replaced by “higher”. The fold factor cannot be obtained as we used the Kruskal-Wallis test which provided the analysis only with average ranks as follow: Complete remission = 17.9, Partial remission = 39.93 and Non-remission = 56.88. The choice of the Kruskal-Wallis test was dictated by the non-Gaussian distribution of the renal PLA2R expression.

Comment 2: In the revised manuscript, from out of the 8 references cited in the introduction section, 5 are original works [3, 5-8]. We did not intend to link Treg decrease to PLA2R expression but to state that both can trigger anti-PLA2R antibody production. Nevertheless, some authors hypothesized that a constant low level of soluble PLA2R maintains peripheral tolerance by controlling Treg function which inhibits anti-PLA2R antibody production by autoreactive B cells [Van de Logt AE, Fresquet M, Wetzels JF, Brenchley P. The anti-PLA2R antibody in membranous nephropathy: what we know and what remains a decade after its discovery. Kidney Int 2019; 96(6): 1292-1302. doi: 10.1016/j.kint.2019.07.014]. In order to make the idea clearer we separated it in two distinct sentences.

Comment 3: We agree that the autoimmune phase could be triggered outside the kidney and we modified the introduction and the discussion sections as recommended by the reviewer #1. In the revised manuscript, we replaced “The current pathophysiological hypothesis stipulates” by “Some authors hypothesized that” at the beginning of the paragraph. We also added that conformational changes and/or increased expression of PLA2R outside the kidney could induce anti-PLA2R antibody production, and in this regard the lungs might be relevant. Both ideas are now supported with appropriate citation.

Comment 4: The study of “Hoxha E, et al” is now cited in the introduction section.

Comment 5: The treatment regimen is now recorded in Table 1 and in a new paragraph in subjects’ section. The treatment of PMN patients was based on the KDIGO guidelines. The combination of prednisone/hydroxychloroquine was used to treat the SLE patients. Twenty-five patients had spontaneous complete remission and were not treated (this data was added to Table 1). Even if complete remission was more frequent in non-nephrotic patients (46.2% vs. 33.3%) the difference failed to reach significance:

The follow-up seems short because the recruitment date ranged from June 2017 to April 2019 and we enrolled only newly diagnosed patients (prospective study).

The SLE patients were used as controls because they had a secondary membranous nephritis and some previous studies reported the presence of anti-PLA2R antibody in some SLE patients. Therefore, in order to better assess sensitivity and specificity of anti-PLA2R antibody measurement, we used SLE patients as the healthy group might be insufficient to obtain accurate results. We did not measure anti-PLA2R antibody in TIN patients because no previous study reported the presence of this autoantibody and glomeruli are not involved in TIN.

TIN patients were chosen as controls for mRNA analysis because they have theoretically non-injured glomeruli with a normal expression of PLA2R which is not the case for patients with membranous lupus nephritis. Moreover, only TIN patients (n=20) biopsies were available and we could not obtain enough usable biopsies for mRNA extraction from SLE patients. 

Comment 6: We analyzed the rs4664308 SNP because it is considered as the most relevant polymorphism inside the PLA2R gene along with the rs2187668 SNP of HLA-DQA1. The strong linkage disequilibrium between rs4664308 and rs3749117 was described in European cohorts [Stanescu HC, Arcos-Burgos M, Medlar A, Bockenhauer D, Kottgen A, Dragomirescu L, et al. Risk HLA-DQA1 and PLA2R1 alleles in idiopathic membranous nephropathy. N Engl J Med 2011; 364(7): 616–26. doi: 10.1056/nejmoa1009742]. The linkage disequilibrium between two or more alleles varies ethnically and it would be interesting to analyze the rs3749117 in our patients in the future as indicated by reviewer #1.

Comment 7:

- At least 3 biopsy slices were used for RNA extraction for each patient. Reverse transcription for cDNA synthesis was made in one step together with the specific amplification of the target or the housekeeping genes. For cDNA synthesis we used the “Rotor-Gene SYBR Green RT-PCR master mix” (Qiagen) and random primers (48190-011, InvitrogenTM). The mix used for reverse transcription and cDNA amplification was as follow: 1) Rotor-Gene SYBR Green RT-PCR Master Mix (2X): 12.5 µl, 2) Rotor-Gene RT-Mix: 0.25 µl, 3) Random primers (150 ng/µl): 1.25 µl, 4) Extracted RNA (concentration: 2 to 4 ng/µl): 5 µl, 5) Specific primers Hs00234853_m1 (PLA2R) or Hs00922492_m1 (Podocin): 1.25 µl and 6): RNase free water: 4.75 µl. Thermal cycling was performed with an initial reverse transcription step at 55°C for 10 minutes, an activation step at 95°C for 5 minutes and then 40 cycles of denaturation at 95°C for 5 seconds and annealing/extension at 60°C for 10 seconds. We added these details to the methods section.

- We used podocin as reference gene in order to replicate the study of Hoxha E, et al (doi: 10.1038/ki.2012.209) in which the authors used the podocin as a housekeeping gene. we also compared the CT-values of podocin between TIN patients and PMN patients and there was no significant difference (p=0.284).

Thus, as shown there was no decrease of podocin expression in our PMN patients. Then we tested the correlation between CT-values of podocin and 24h-proteinuria, and there was no significant correlation, Spearman Rho = -0.112, p=0.362.

After that we tested the association between baseline (at the moment of kidney biopsy) nephrotic syndrome and podocin CT-values and again there was no significant association, p=0.865.

Since the CT-values did not vary between PMN and TIN patients and were not influenced by 24h-proteinuria or the presence of nephrotic syndrome we concluded that podocin was a good housekeeping gene in our study. Moreover, we have already compared the podocin and the RNA 18S Ct-values in 26 patients. There was a significant correlation between Ct-values of podocin and RNA 18S, Spearman Rho = 0.731, p=2.2E-5.

- We agree with reviewer #1 that immunostaining of biopsies and/or western blotting would be interesting in the future.

Comment 8: As recommended by reviewer #1 we added dot plots to figures 1, 2, 3 and 5. Additional box plot (1b) and dot plot (1c) were made to show separately and quantitatively anti-PLA2R Ab distribution and values in positive and negative PMN patients for anti-PLA2R Ab. We also added the threshold value (20 RU/ml) to figure 1 in box plots (1(a) and 1(c)) and in dot plots (1(b) and 1(d)). In figure 3, a dot plot (3b) showing PLA2R mRNA level in patients with positive anti-PLA2R Ab vs. negative vs. TIN patients. For figure 4, all PMN patients (positive and negative for anti-PLA2R Ab) were plotted for correlation with renal PLA2R level.

Comment 9: PLA2R mRNA levels and baseline 24h-proteinuria were not correlated, Spearman Rho = -0.069, p=0.577. This data was added to results section.

Anti-PLA2R Ab levels, and baseline 24h-proteinuria were not correlated, Spearman Rho = -0.080, p=0.515. We also added this data to results section.

Podocin CT values were not correlated to baseline 24h-proteinuria, Spearman Rho = -0.112, p=0.362.

Comment 10: Percentages for males vs. females are now compared vertically in Table 2. In table 1 there are no percentages comparing males vs. females but a descriptive data on sex-ratio for the PMN group. We indicated in tables 1, 2 and 4 when the parameters were measured (baseline or at diagnosis).

Comment 11: We analyzed the correlation between baseline 24h-proteinuria and PMN outcome and there was no significant association, p=0.422. We added this data to the results section.

A multivariable analysis was performed using the following covariables: baseline 24h-protéinuria, rs4664308 SNP, the presence of nephrotic syndrome at diagnosis, the presence of kidney failure at diagnosis, and the association between anti-PLA2R Ab (qualitatively and quantitatively) and PMN outcome remained statistically significant. This data was added to results section.

Also, another multivariable analysis was performed the association between PLA2R mRNA level and PMN outcome remained statistically significant. This data was added to results section.

Comment 12: Yes, previous study showed that age of PMN patients anti-PLA2R-positive vs. anti-PLA2R-negative was similar and we stated that our peculiar finding has not been noted in previous reports. We added the recommended citation by reviewer #1.

Minor comments

Comment 1: We replaced all “wild” with “wild-type” in the manuscript.

Comment 2: All “analytic results” have been deleted from the text.

Comment 3: It is the fold-change comparatively to a calibrator (biopsy from a kidney living donor). For each sample a relative expression of PLA2R to podocin was calculated. Then a fold-change for relative PLA2R expression was obtained from the calibrator.

Comment 4: We deleted or replaced most of “besides” in the text.

Comment 5: the legend for the Y axis in Figure 5 has been corrected.

Reviewer #2:

Major comments

Comment 1: TIN patients were chosen as controls for mRNA analysis because they have theoretically non-injured glomeruli with a normal expression of PLA2R which is not the case for patients with membranous lupus nephritis. Moreover, only TIN patients (n=20) biopsies were available and we could not obtain enough usable biopsies for mRNA extraction from SLE patients or kidney living donors.

Comment 2: Yes, all SLE patients had pure class V membranous lupus nephropathy. We specified this detail in subjects’ section.

Comment 3: PMN diagnosis was confirmed by light microscopy analysis of renal biopsies for all patients without staining for PLA2R or IgG4. We did not perform immunochemistry or direct immunofluorescence. We specified in the subjects’ section that PMN diagnosis was made on the basis of light microscopy. We also specified that all patients were screened for secondary membranous nephritis etiologies.

Comment 4: In this prospective study the anti-PLA2R Ab levels are baseline and in the same time at the time of biopsies. We specified it in the table 2.

Comment 5: In choosing a cut-off value of 239.95 RU/ml, the sensitivity and the specificity for non-remission prediction were respectively 100% and 87.5%. We updated this new data in the results and discussion sections. Thus, comparatively to the former cut-off value of 372.1 RU/ml, the sensitivity increased from 66.7% to 100% and the specificity decreased from 91.1% to 87.5%.

Comment 6: The lower mRNA level association with CKD can be explained the destruction of podocytes concomitant with the onset of renal failure. As the renal PLA2R expression is significantly correlated to anti-PLA2R Ab levels, and since this autoantibody level is associated with the non-remission outcome, this apparently paradoxical finding can be explained by the fact that lower mRNA levels induce less anti-PLA2R Ab and could be predictive of a better outcome despite its association with a baseline renal failure. This explanation has been added to the discussion section.

Comment 7: Since we screened SLE patients with secondary membranous nephritis we discussed our findings comparatively to previous studies in which a rare presence of anti-PLA2R Ab was reported. Consequently, we added possible explanations for the presence of this autoantibody in secondary membranous nephritides for which opinions differ, in fact some authors suggested that the presence of anti-PLA2R antibodies in patients with SMN might result from the co-incidental simultaneous development of PMN and a systemic disease, such as SLE or a malignancy. We felt that it is important to discuss the occurrence of anti-PLA2R Ab in secondary membranous nephritides even if it was negative in our SLE patients.

Comment 8: We changed ‘autoantibody’ to ‘immunological’ and as the immunological remission is well established, we changed ‘could be’ to ‘is’.

Minor comments

Comment 1: We corrected the errors mentioned by reviewer #2 and an academic English teacher has reviewed and corrected the manuscript.

Comment 2: We change p-values to <0.0001 when small.

Comment 3: We clarified that the rs4664308*G allele is protective.

Comment 4: We changed CKD classification into stage I to V in the table 1.

---

## [Decision Letter · Decision Letter 1]

4 Sep 2020

PONE-D-20-17580R1

PLA2R antibody, PLA2R rs4664308 polymorphism and PLA2R mRNA levels in Tunisian patients with primary membranous nephritis

PLOS ONE

Dear Dr. Dhaouadi,

Thank you for submitting your manuscript to PLOS ONE. After careful consideration, we feel that it has merit but does not fully meet PLOS ONE’s publication criteria as it currently stands. Therefore, we invite you to submit a revised version of the manuscript that addresses the points raised during the review process.

**The revised manuscript is definitely improved. However, there are few points that remain to be addressed. To mention some of them, i) unclear the issue of the fold-factor, which is not dependent on the statistical test used; ii) need to delete all the sentences in the introduction on the etiology or factors associated to the production of anti-PLA2R1, as this is not related to the measurements performed in the present manuscript; iii) there is no need to discuss the level of PLA2R mRNA in regard to etiology of production of antibodies; iv) need to cite the ‘Hoxha et al’ study in the context of the present study; v) need to discuss the discrepancy of the present study with that of Hoxha et al; vi) need to acknowledge in the discussion that some patients were non-nephrotic at baseline; vii) need to indicate in the manuscript what covariables are used in the authors’ multivariable models.**

We look forward to receiving your revised manuscript.

Kind regards,

Giuseppe Remuzzi

Academic Editor

PLOS ONE

Reviewers' comments:

Reviewer's Responses to Questions

**Comments to the Author**

1. If the authors have adequately addressed your comments raised in a previous round of review and you feel that this manuscript is now acceptable for publication, you may indicate that here to bypass the “Comments to the Author” section, enter your conflict of interest statement in the “Confidential to Editor” section, and submit your "Accept" recommendation.

Reviewer #1: (No Response)

Reviewer #2: All comments have been addressed

2. Is the manuscript technically sound, and do the data support the conclusions?

Reviewer #1: Partly

Reviewer #2: Yes

3. Has the statistical analysis been performed appropriately and rigorously? 

Reviewer #1: Yes

Reviewer #2: Yes

4. Have the authors made all data underlying the findings in their manuscript fully available?

Reviewer #1: Yes

Reviewer #2: Yes

5. Is the manuscript presented in an intelligible fashion and written in standard English?

Reviewer #1: Yes

Reviewer #2: Yes

6. Review Comments to the Author

Reviewer #1: I thank the authors for their reply to my previous comments. Please see below my further comments over the Authors' response.

Reviewer #1:

Major comments

Comment 1: We provided the abstract with information on techniques used and we specified that PMN patients were analyzed for renal PLA2R expression. “increased” was replaced by “higher”. The fold factor cannot be obtained as we used the Kruskal-Wallis test which provided the analysis only with average ranks as follow: Complete remission = 17.9, Partial remission = 39.93 and Non-remission = 56.88. The choice of the Kruskal-Wallis test was dictated by the non-Gaussian distribution of the renal PLA2R expression.

Reviewer #1: I don't understand. The fold-factor is not dependent on the statistical test used. It is simply the ratio between mean values of expression of PLA2R1.

Comment 2: In the revised manuscript, from out of the 8 references cited in the introduction section, 5 are original works [3, 5-8]. We did not intend to link Treg decrease to PLA2R expression but to state that both can trigger anti-PLA2R antibody production. Nevertheless, some authors hypothesized that a constant low level of soluble PLA2R maintains peripheral tolerance by controlling Treg function which inhibits anti-PLA2R antibody production by autoreactive B cells [Van de Logt AE, Fresquet M, Wetzels JF, Brenchley P. The anti-PLA2R antibody in membranous nephropathy: what we know and what remains a decade after its discovery. Kidney Int 2019; 96(6): 1292-1302. doi: 10.1016/j.kint.2019.07.014]. In order to make the idea clearer we separated it in two distinct sentences.

Reviewer #1: In my opinion, all the sentences in the introduction on the etiology or factors associated to the production of anti-PLA2R1 can be deleted, as this is not related to the measurements performed this manuscript. No Treg number is measured and the higher increase of PLA2R mRNA is likely due to the disease activity (ie formation of immune complexes), not linked to etiology leading to the production of anti-PLA2R1 antibodies.

Comment 3: We agree that the autoimmune phase could be triggered outside the kidney and we modified the introduction and the discussion sections as recommended by the reviewer #1. In the revised manuscript, we replaced “The current pathophysiological hypothesis stipulates” by “Some authors hypothesized that” at the beginning of the paragraph. We also added that conformational changes and/or increased expression of PLA2R outside the kidney could induce anti-PLA2R antibody production, and in this regard the lungs might be relevant. Both ideas are now supported with appropriate citation.

Reviewer #1: The authors have only partially replied to my suggestion and the sentences are still vague. Again, I don't see the need to discuss the level of PLA2R mRNA in regard to etiology of production of antibodies. What is important here the correlation between mRNA levels and antibodies, which suggest some triggering of the immune response, not necessarily linked to the real initiation of the auoimmune response, with loss of tolerance, etc.

Comment 4: The study of “Hoxha E, et al” is now cited in the introduction section.

Reviewer #1:The way the study is cited is not put in the context of this study. Please rephrase better. In the discussion, they should clearly discuss why the qPCR authors' results are completely different: Hoxha et al saw no change in mRNA level while this study shows a dramatic increase. Please discuss carefully why this discrepancy.

Comment 5: The treatment regimen is now recorded in Table 1 and in a new paragraph in subjects’ section. The treatment of PMN patients was based on the KDIGO guidelines. The combination of prednisone/hydroxychloroquine was used to treat the SLE patients. Twenty-five patients had spontaneous complete remission and were not treated (this data was added to Table 1). Even if complete remission was more frequent in non-nephrotic patients (46.2% vs. 33.3%) the difference failed to reach significance:

Reviewer #1: Thank you for providing information about treatment. The authors should still acknowledge in the discussion that some patients were non-nephrotic at baseline, which is a limitation of their study when considering clinical outcome and biomarkers.

The follow-up seems short because the recruitment date ranged from June 2017 to April 2019 and we enrolled only newly diagnosed patients (prospective study).

The SLE patients were used as controls because they had a secondary membranous nephritis and some previous studies reported the presence of anti-PLA2R antibody in some SLE patients. Therefore, in order to better assess sensitivity and specificity of anti-PLA2R antibody measurement, we used SLE patients as the healthy group might be insufficient to obtain accurate results. We did not measure anti-PLA2R antibody in TIN patients because no previous study reported the presence of this autoantibody and glomeruli are not involved in TIN.

TIN patients were chosen as controls for mRNA analysis because they have theoretically non-injured glomeruli with a normal expression of PLA2R which is not the case for patients with membranous lupus nephritis. Moreover, only TIN patients (n=20) biopsies were available and we could not obtain enough usable biopsies for mRNA extraction from SLE patients.

Comment 6: We analyzed the rs4664308 SNP because it is considered as the most relevant polymorphism inside the PLA2R gene along with the rs2187668 SNP of HLA-DQA1. The strong linkage disequilibrium between rs4664308 and rs3749117 was described in European cohorts [Stanescu HC, Arcos-Burgos M, Medlar A, Bockenhauer D, Kottgen A, Dragomirescu L, et al. Risk HLA-DQA1 and PLA2R1 alleles in idiopathic membranous nephropathy. N Engl J Med 2011; 364(7): 616–26. doi: 10.1056/nejmoa1009742]. The linkage disequilibrium between two or more alleles varies ethnically and it would be interesting to analyze the rs3749117 in our patients in the future as indicated by reviewer #1.

Reviewer #1: OK, you should indicate this in the main text.

Comment 7:

- At least 3 biopsy slices were used for RNA extraction for each patient. Reverse transcription for cDNA synthesis was made in one step together with the specific amplification of the target or the housekeeping genes. For cDNA synthesis we used the “Rotor-Gene SYBR Green RT-PCR master mix” (Qiagen) and random primers (48190-011, InvitrogenTM). The mix used for reverse transcription and cDNA amplification was as follow: 1) Rotor-Gene SYBR Green RT-PCR Master Mix (2X): 12.5 µl, 2) Rotor-Gene RT-Mix: 0.25 µl, 3) Random primers (150 ng/µl): 1.25 µl, 4) Extracted RNA (concentration: 2 to 4 ng/µl): 5 µl, 5) Specific primers Hs00234853_m1 (PLA2R) or Hs00922492_m1 (Podocin): 1.25 µl and 6): RNase free water: 4.75 µl. Thermal cycling was performed with an initial reverse transcription step at 55°C for 10 minutes, an activation step at 95°C for 5 minutes and then 40 cycles of denaturation at 95°C for 5 seconds and annealing/extension at 60°C for 10 seconds. We added these details to the methods section.

Reviewer #1: OK thanks for this.

- We used podocin as reference gene in order to replicate the study of Hoxha E, et al (doi: 10.1038/ki.2012.209) in which the authors used the podocin as a housekeeping gene. we also compared the CT-values of podocin between TIN patients and PMN patients and there was no significant difference (p=0.284).

Thus, as shown there was no decrease of podocin expression in our PMN patients. Then we tested the correlation between CT-values of podocin and 24h-proteinuria, and there was no significant correlation, Spearman Rho = -0.112, p=0.362.

After that we tested the association between baseline (at the moment of kidney biopsy) nephrotic syndrome and podocin CT-values and again there was no significant association, p=0.865.

Since the CT-values did not vary between PMN and TIN patients and were not influenced by 24h-proteinuria or the presence of nephrotic syndrome we concluded that podocin was a good housekeeping gene in our study. Moreover, we have already compared the podocin and the RNA 18S Ct-values in 26 patients. There was a significant correlation between Ct-values of podocin and RNA 18S, Spearman Rho = 0.731, p=2.2E-5.

Reviewer #1: Thank you for the detailed analyses. I recommend to add the key points in the manuscript to support the finding that podocin is a bona fide housekkeping gene. However, the authors should explain or provide an hypothesis of why their findings are different from the previous study by Hoxha et al., using the same methodology.

- We agree with reviewer #1 that immunostaining of biopsies and/or western blotting would be interesting in the future.

Reviewer #1: I think this is mandatory to confirm the qPCR findings and improve the quality of the manuscript. If not done, the authors should clearly indicate in the discussion that this has to be done in the future to confirm the novel and different qPCR findings, as compared to Hoxha' study.

Comment 8: As recommended by reviewer #1 we added dot plots to figures 1, 2, 3 and 5. Additional box plot (1b) and dot plot (1c) were made to show separately and quantitatively anti-PLA2R Ab distribution and values in positive and negative PMN patients for anti-PLA2R Ab. We also added the threshold value (20 RU/ml) to figure 1 in box plots (1(a) and 1(c)) and in dot plots (1(b) and 1(d)). In figure 3, a dot plot (3b) showing PLA2R mRNA level in patients with positive anti-PLA2R Ab vs. negative vs. TIN patients. For figure 4, all PMN patients (positive and negative for anti-PLA2R Ab) were plotted for correlation with renal PLA2R level.

Reviewer #1: Please add the last above statement in the manuscript.

Comment 9: PLA2R mRNA levels and baseline 24h-proteinuria were not correlated, Spearman Rho = -0.069, p=0.577. This data was added to results section.

Anti-PLA2R Ab levels, and baseline 24h-proteinuria were not correlated, Spearman Rho = -0.080, p=0.515. We also added this data to results section.

Podocin CT values were not correlated to baseline 24h-proteinuria, Spearman Rho = -0.112, p=0.362.

Reviewer #1: Please add the last above statement in the manuscript.

Comment 10: Percentages for males vs. females are now compared vertically in Table 2. In table 1 there are no percentages comparing males vs. females but a descriptive data on sex-ratio for the PMN group. We indicated in tables 1, 2 and 4 when the parameters were measured (baseline or at diagnosis).

Reviewer #1: Thanks.

Comment 11: We analyzed the correlation between baseline 24h-proteinuria and PMN outcome and there was no significant association, p=0.422. We added this data to the results section.

Reviewer #1: This is uncommon as proteinuria is a well-known clinical biomarker (Thompson, A., Cattran, D. C., Blank, M., and Nachman, P. H. (2015) Complete and Partial Remission as Surrogate End Points in Membranous Nephropathy, J Am Soc Nephrol 26, 2930-7). What would be the reason: non-nephrotic patients in this cohort, cohort size?

A multivariable analysis was performed using the following covariables: baseline 24h-protéinuria, rs4664308 SNP, the presence of nephrotic syndrome at diagnosis, the presence of kidney failure at diagnosis, and the association between anti-PLA2R Ab (qualitatively and quantitatively) and PMN outcome remained statistically significant. This data was added to results section.

Also, another multivariable analysis was performed the association between PLA2R mRNA level and PMN outcome remained statistically significant. This data was added to results section.

Reviewer #1: The authors should indicate in the manuscript what covariables are used in their multivariable models.

Comment 12: Yes, previous study showed that age of PMN patients anti-PLA2R-positive vs. anti-PLA2R-negative was similar and we stated that our peculiar finding has not been noted in previous reports. We added the recommended citation by reviewer #1.

Reviewer #1: Thanks.

Minor comments

Comment 1: We replaced all “wild” with “wild-type” in the manuscript.

Reviewer #1: Thanks.

Comment 2: All “analytic results” have been deleted from the text.

Reviewer #1: Thanks.

Comment 3: It is the fold-change comparatively to a calibrator (biopsy from a kidney living donor). For each sample a relative expression of PLA2R to podocin was calculated. Then a fold-change for relative PLA2R expression was obtained from the calibrator.

Reviewer #1: Please indicate this in the figure legend.

Comment 4: We deleted or replaced most of “besides” in the text.

Reviewer #1: Thanks.

Comment 5: the legend for the Y axis in Figure 5 has been corrected.

Reviewer #1: I am talking about figure 4 where the legend ("anti-PLA2R Ab level (RU/mL)") of the Y axis is not indicated on the left side but on the top of the figure (as a title, which is misleading).

Reviewer #2: (No Response)

7. PLOS authors have the option to publish the peer review history of their article (what does this mean?). If published, this will include your full peer review and any attached files.

Reviewer #1: No

Reviewer #2: No

---

## [Author Response · Author response to Decision Letter 1]

7 Sep 2020

Review comments to the author

Reviewer #1:

Major comments

Comment 1: We added the fold-factors of PLA2R expression in non-remission vs. partial remission and vs. complete remission as requested by the reviewer #1.

Comment 2: We deleted all the sentences related to the etiology of anti-PLA2R Ab production.

Comment 3: As requested by reviewer #1 we replaced the link between PLA2R expression and autoimmune initiation by the correlation with anti-PLA2R production. The new sentence is: Some authors hypothesized that a renal overexpression of PLA2R could trigger the immune response and be correlated to anti-PLA2R Ab production. We also deleted the last sentence of the paragraph in which the PLA2R expression in lung could induce anti-PLA2R Ab.

Comment 4: We added this sentence in the introduction: Of note, in the study of Hoxha et al [8], the authors did not report any significant difference in PLA2R expression between patients with enhanced PLA2R staining and those with no enhanced staining.

In the discussion we added this sentence: Moreover, in the study of Hoxha et al [8], the compared groups included only MN patients while in our study the controls were TIN patients which could explain the discrepancy between the results. The following sentence explains why possibly an enhanced PLA2R staining is not necessarily correlated to PLA2R mRNA because of post-transcriptional increase.

Comment 5: We added this sentence while discussing PLA2R Ab levels correlation with PMN outcome: Even there is a limitation due to the presence of some non-nephrotic patients at baseline, …

We also added this sentence while discussing the association between PLA2R expression and PMN outcome: Nevertheless, the presence of some non-nephrotic patients at baseline remains a limitation in this regard.

Comment 6: As requested by reviewer #1, we added this sentence in the main text: Nevertheless, the linkage disequilibrium between two or more alleles varies ethnically and needs to be assessed in independent cohorts.

Comment 7:

We added this paragraph in the main text: We used podocin as a housekeeping gene in order to replicate the study of Hoxha E, et al [8]. Comparison of the CT-values of podocin between TIN and PMN patients showed no significant difference (p=0.284). Moreover, we tested the correlation between CT-values of podocin and 24h-proteinuria, and there was no significant correlation, Spearman Rho = -0.112, p=0.362. Likewise, podocin CT-values were not associated to nephrotic syndrome at the moment of kidney biopsy, p=0.865. Since the CT-values of podocin did not vary between PMN and TIN patients and were not influenced by 24h-proteinuria or the presence of nephrotic syndrome we concluded that podocin was a good housekeeping gene in our study. Moreover, comparison of podocin and the RNA 18S Ct-values in 26 PMN patients showed a significant correlation; Spearman Rho = 0.731, p=2.2E-5.

We discussed the discrepancy with the study of Hoxha E et al as requested in comment 4 and 7.

Comment 8: We added this sentence to Fig 4 legend: All PMN patients (positive and negative for anti-PLA2R Ab) were plotted for correlation with renal PLA2R level.

Comment 9: We already added this data “Podocin CT values were not correlated to baseline 24h-proteinuria, Spearman Rho = -0.112, p=0.362” as requested in comment 7 and comment 9

Comment 10: No request

Comment 11: We added this paragraph to the discussion section: In this study, 68 PMN patients were enrolled. Only 42 (61.8%) patients were nephrotic at the moment of diagnosis. Nevertheless, only 12 (17.6%) PMN patients evolved to a non-remission. Of note, PMN outcome was not influenced by baseline 24-hour proteinuria, p=0.422. This is an uncommon finding as proteinuria is generally recognized as a major prognostic factor [14]. This issue could be due to the generally good prognosis of our patients and the small size of our cohort. 

We also specified in the results section the covariable used for the multivariable analysis.

Comment 12: No request

Minor comments

Comment 1: No request

Comment 2: No request

Comment 3: we added details of Fold-change calculation to fig 3 legend.

Comment 4: No request

Comment 5: the legend for the Y axis in Figure 4 has been corrected, the “anti-PLA2R Ab level” is on the left side now.

Reviewer #2: No request

---

## [Decision Letter · Decision Letter 2]

18 Sep 2020

PLA2R antibody, PLA2R rs4664308 polymorphism and PLA2R mRNA levels in Tunisian patients with primary membranous nephritis

PONE-D-20-17580R2

Dear Dr. Dhaouadi,

We’re pleased to inform you that your manuscript has been judged scientifically suitable for publication and will be formally accepted for publication once it meets all outstanding technical requirements.

**The re-revised version of the manuscript is improved. The authors have properly addressed all the remaining comments.**

Kind regards,

Giuseppe Remuzzi

Academic Editor

PLOS ONE

Additional Editor Comments (optional):

Reviewers' comments:

Reviewer's Responses to Questions

**Comments to the Author**

1. If the authors have adequately addressed your comments raised in a previous round of review and you feel that this manuscript is now acceptable for publication, you may indicate that here to bypass the “Comments to the Author” section, enter your conflict of interest statement in the “Confidential to Editor” section, and submit your "Accept" recommendation.

Reviewer #1: All comments have been addressed

2. Is the manuscript technically sound, and do the data support the conclusions?

Reviewer #1: Yes

3. Has the statistical analysis been performed appropriately and rigorously? 

Reviewer #1: Yes

4. Have the authors made all data underlying the findings in their manuscript fully available?

Reviewer #1: Yes

5. Is the manuscript presented in an intelligible fashion and written in standard English?

Reviewer #1: Yes

6. Review Comments to the Author

Reviewer #1: The authors have now considered all of my comments. The differences in mRNA amount for PLA2R1 with the previous study by Hoxha is the interesting point of this study. To be confirmed..

7. PLOS authors have the option to publish the peer review history of their article (what does this mean?). If published, this will include your full peer review and any attached files.

Reviewer #1: No

---

## [Editor Report · Acceptance letter]

22 Sep 2020

PONE-D-20-17580R2

PLA2R antibody, PLA2R rs4664308 polymorphism and PLA2R mRNA levels in Tunisian patients with primary membranous nephritis

Dear Dr. Dhaouadi:

I'm pleased to inform you that your manuscript has been deemed suitable for publication in PLOS ONE. Congratulations! Your manuscript is now with our production department.

Kind regards,

on behalf of

Prof. Giuseppe Remuzzi 

Academic Editor

PLOS ONE